# Enlargement of the Nerve Fibers of Silenced Lumbosacral Motoneurons in Cats

**DOI:** 10.3390/biomedicines10102622

**Published:** 2022-10-18

**Authors:** Tessa Gordon, Lynn Eldridge, Saljae Aurora

**Affiliations:** 1Department of Surgery, Division of Plastic Reconstructive Surgery, University of Toronto, Toronto, ON M5G 1X8, Canada; 2Department of Physiology, School of Medicine, University of California, Los Angeles (UCLA), Los Angeles, CA 90024, USA; 3Department of Oral Health Sciences, Faculty of Dentistry, University of British Columbia, 2199 Westbrook Mall, Vancouver, BC V6T 1Z3, Canada

**Keywords:** motoneuron plasticity, neuromuscular activity, motor nerves, peripheral nerve size

## Abstract

Whether neuromuscular activity influences the size of motor nerves is controversial. All neuromuscular activity in cat hindlimbs was eliminated by spinal cord isolation (SCI), namely, spinal cord transection above and below the medial gastrocnemius (MG) and soleus (SOL) motoneuron pools and L5-S3 dorsal root transection. MG, SOL and sural (SUR) nerves were removed for size measurements, eight months after SCI surgery and from age-matched control cats. Nerve fiber number, the linear relationship between axon size and myelin thickness, and the bimodal distributions of nerve fiber area and diameter were maintained in all three nerves after SCI. The distributions of myelinated sensory fibers were unchanged in SUR nerves in contrast to the myelinated motor fibers in the MG and SOL nerves that were significantly larger. These findings provide evidence that all lumbar motoneurons survive SCI and that their nerve fibers enlarge. Thus, motor nerve fiber size in addition to the properties of the motoneurons and their muscle fibers is dynamic, responding to neuromuscular activity.

## 1. Introduction

Functional specialization of peripheral nerves by size has been implied in several systems. An interesting example is the frequency coding in the cochlea nerve with appropriate matching of cochlea nerve fiber diameters to their firing frequencies [1]. Another example is the matching of motoneuron size and motor nerve conduction velocity to the force output of the muscle fibers supplied by the motoneurons, force increasing with the size-the so-called Size Principle [2,3,4]. This matching is re-established once regenerating motor nerves make functional contact with muscle fibers after Wallerian degeneration and regeneration of cut and surgically repaired hindlimb nerves [5,6].

The size of motor nerve fibers depends critically on contact with their targets. This was shown by the decline in the conduction velocities recorded proximal to the site of their transection and surgical repair that is followed by their full recovery once the regenerating fibers made functional connections with their denervated muscle fibers [5,6,7,8]. That the dimensions of motor nerve fibers were unaltered when muscles were partially denervated by transection of some of their nerve supply [9] argues that motor nerve fiber size does not depend on how many muscle fibers are supplied by each nerve.

Peripheral nerve fiber caliber is also affected by neuromuscular activity [10]. Increased levels of neuromuscular activity by intense muscle exercise or contralateral limb immobilization, conditions which are frequently associated with increased muscle bulk, resulted in significant reductions in nerve fiber diameters [11,12,13,14,15] rather than the increases that might be expected from the earlier suggestion of Edds [16]. However, data on the influence of activity on nerve fiber size are conflicting, probably due to the wide variability in the levels of neuromuscular activity from study to study which, in turn, varies with the model of hyper-and hypo-activities employed (reviewed by [17]). Nonetheless, the consistent findings that nerve fiber size and/or conduction velocities decreased under conditions of strenuous exercise [18] and after high levels of daily, low frequency electrical stimulation of the motor nerve [19], and that the fiber sizes increased or remained unchanged with less frequent activity [12,13,14], muscle overload [16], and conditions of hypoactivity [14,15,20], suggest that nerve fiber size may be inversely related to the total activity of the motoneurons rather than directly related to muscle bulk. However, no change in size of identified rat soleus and tibialis anterior motoneurons or of cat plantaris motoneurons was reported one and three months after spinal cord isolation (SCI) and muscle overload, respectively [21,22].

In this study, we have dissociated the effects of motoneuronal activity and muscle bulk on nerve fiber size to test the hypothesis that synaptic input into the motoneuron pool rather than muscle bulk is the determinant of motor nerve size. To do so, we silenced motoneurons for a prolonged period of 8 months, by their deafferentation and removal of supra- and infra-spinal input, as described previously [23,24,25,26]. After this surgery, there is almost no detectable electromyographic activity in the atrophic hindlimb muscles [23] but low frequency firing of a single motor unit was detected in a few cat hindlimbs [26].

## 2. Materials and Methods

### 2.1. Animals

A total of 14 domestic, long-tailed cats (3–5 kg, male and female) were used in the experiments. They were procured from the Los Angeles and Ventura County Animal Shelters. In 11 cats, the spinal cord was transected rostral to L5 and at or below S3 segments and the L5-L7 and S1-S3 dorsal roots were cut proximal to their cell bodies in the dorsal root ganglia (Figure 1). The surgical procedure, often referred to as spinal cord isolation (SCI), was performed in the laboratory of Dr. Lynn Eldridge at the University of California at Los Angeles, under the American Association for Accreditation of Laboratory Animal Care (AAALAC) certification. The detailed description of the surgeries carried out under meticulous supervision and the subsequent health care program, are published [23] and summarized below.

### 2.2. Sterile Surgery

Intramuscular antibiotic (Bicillin 10 mg/Kg) was administered to all experimental cats one hour before sterile surgery to reduce the risk of infection. No postsurgical infection was recorded following these surgeries. The cats were anesthetized with an intraperitoneal (i.p.) injection of sodium pentobarbital (Somnotol 40 mg/kg). The rectal temperatures were monitored during the 3–4 h surgery.

A midline dorsal incision through the skin and fascia was made from the L3 to S2, and muscle scraped from the bones to perform a small narrow laminectomy of ~2 mm in diameter. Under 6× magnification on a Zeiss operating microscope, the laminectomy was widened to the diameter of the underlying cord to transect the cord at the caudal border of the L4 segment and caudal to segment S3, making sure to spare the ventral artery for blood supply to the isolated ventral spinal cord (Figure 1A). The isolated spinal cord between the two transection sites contains the lumbosacral motoneurons that innervate calf muscles, including medial gastrocnemius (MG) and soleus (SOL). Their numbers were normal, and their cell bodies retained their normal appearance for several years [27]. The narrow laminectomy between the transection sites also preserved vertebral structures and prevented accidental cord stimulation by pressure on the back or by any movements of the cat.

All the dorsal roots of the isolated spinal cord were exposed by rotating the cord on its central axis to left and then right sides with gentle traction. They were cut intradurally close to their junctions with the ventral roots, taking care not to manipulate the latter roots and to ensure their integrity (Figure 1B). This surgery retained the sensory neuronal cell bodies in the dorsal root ganglia with their axons in the paralyzed hindlimbs. The dura was left open, and the exposed cord covered with Gelfilm to protect it and to prevent any indirect pressure by edema. The muscles on either side of the spinal column were sutured together, and the facia and the skin incisions closed.

**Figure 1 biomedicines-10-02622-f001:**
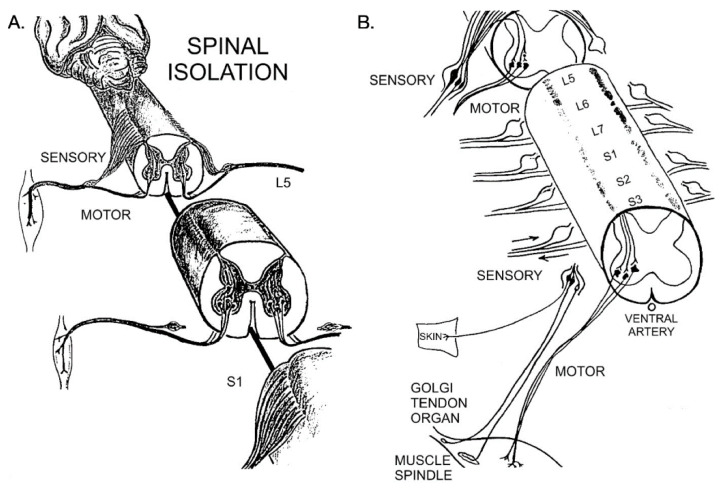
Diagrammatic representation of the surgery of spinal cord isolation as first described for dogs in 1937 [28] and for cats in 1981 [25]. (**A**) The lumbosacral spinal cord that contains the motoneurons that innervate the medial gastrocnemius and the soleus muscles was isolated from synaptic input by transecting the spinal cord above and below the location of the motoneurons, leaving the ventral artery intact, and (**B**) cutting the dorsal sensory spinal roots intradurally thereby isolating the sensory neurons in the dorsal root ganglia and their sensory nerve fibers that supply the skin, Golgi tendon organs and muscle spindles. More details are given in the text.

### 2.3. Post-Surgical Care

The cats were placed on a heating pad where they recovered consciousness over an 18–30 h period. Once awake, they were accommodated with two other SCI cats in walk-in 3.1 m^2^ pens where daily replaceable, waffle-patterned cardboard pads that provided cushioning, warmth, and moisture absorption. Each cat was checked daily, and their bladders emptied manually by the Dr. Eldridge or her assistant. Daily manipulation of the paralyzed limbs prevented freezing of the ankle joints.

### 2.4. Histology

#### 2.4.1. Nerve Excision and Preparation

Three weeks and 8 months after the SCI surgery, final acute experiments were performed under pentobarbital anesthesia to record contractile properties [26] Thereafter, muscles were removed for analysis of troponin I isotypes [29], the spinal cord removed for histology [27], and the MG and SOL nerves, 7–16 mm in length, were removed bilaterally ~10 mm proximal to the entry of the nerves into their respective muscles. Sural (SUR) nerves of similar length were excised at the level of the popliteal fossa. The cats were euthanized, and the nerves fixed in 3% glutaraldehyde in 0.1 M phosphate buffer, for immediate courier transport to Edmonton where they were immediately post-fixed in osmium tetroxide for 1 h, dried in ascending alcohols, and embedded in araldite. Cross-sections of 1 µm thick were cut and photographed, at magnifications of 160× and 400×.

#### 2.4.2. Nerve Fiber and Axon Measurements

Enlarged photographs of the cross-sections of the myelinated nerves (Figure 2) were used to number the myelinated fibers with a digitizing tablet attached to a computer. A cursor with a crosshair was used to measure axon perimeters (s) adjacent to the innermost myelin lamellae and the outermost myelin lamella perimeters (S) of all the myelinated nerves. The rectangular co-ordinates of the crosshair position during the tracing process were recorded using a microcomputer (Kaypro 2000). The perimeters and the areas enclosed by them, the axon area (a) and nerve fiber area (A), were calculated from these co-ordinates. The equivalent axon diameter (d_eq_) and equivalent myelin diameter (D_eq_) were calculated from these measurements in the same way from the A, namely D_eq_ = 2/A√π. These measurements provide the most accurate means of determining true fiber size [29,30]. The inner and outer perimeters were used to calculate equivalent myelin thickness (m) where m = (S − s) 2π) [30,31].

### 2.5. Statistics

Regression lines were fitted to the data with the use of least-mean squares criteria and the Pearson product-moment correlation coefficients were calculated according to standard equations for a linear regression [32]. Correlation of X and Y variables was accepted if the slopes of the regression lines were significantly different from zero at *p* < 0.05. Significant differences between pooled cumulative distributions were tested by applying the Kolmogorov–Smirnov test [33]. Differences were significant when *p* < 0.05. *p* values < 0.01 were also given. Mean values ± standard error values are given where appropriate.

## 3. Results

### 3.1. Nerve Fiber Numbers and Dimensions

In the experimental cats, the motoneurons in the lumbosacral spinal cord were subjected to spinal cord isolation (SCI) by section of the spinal cord at L4 and S3 and bilateral section of the L5-7 and S1-3 dorsal roots proximal to the dorsal root ganglia (Figure 1). Typical cross-sections of medial gastrocnemius (MG) and soleus (SOL) nerves from the hindlimbs of a control unoperated cat and an experimental cat are shown in Figure 2. There were no differences detected between the number or size of the myelinated nerve fibers and their connective tissue sheaths in any of the nerves of the control or experimental cats. Their shape changed from the more oblong fiber shape in the control nerves to a rounder shape in the experimental nerve fibers after SCI (Figure 2). Eight months after SCI surgeries, the mean numbers [± standard errors (SE)] of 766 ± 73 and 456 ± 105 myelinated nerve fibers in the MG and SOL nerves, respectively, were not significantly different from 806 ± 106 and 445 ± 42 in the control nerves (*p* > 0.05). They also compared favorably with previously reported numbers [34].

Eight months after SCI, the bimodal distribution of the nerve fiber (axon + myelin) area of the sensory sural (SUR) nerve whose central but not peripheral axons were cut, overlapped with the corresponding distribution at of the control SUR nerve fibers (Figure 3A). The fibers with a peak at ~30 µm^2^ correspond with type II and III sensory fibers and the peak at 100 µm^2^, with the larger type I muscle spindle and tendon afferent fibers (Figure 3A). The equivalent fiber diameters are in accordance with previous measurements [7,35].

The fiber area and equivalent fiber diameter distributions of the SOL (*n* = 9) and MG (*n* = 7) nerves were also bimodal in the hindlimbs of normal and experimental cats, 8 months after SCI (Figure 3B,C). These nerves contain 1a muscle and 1b joint afferent nerve fibers and motor efferent fibers, the first peak attributed to the 1b afferents and the second to 1a afferent and the efferent nerve fibers. The first peaks of the distributions overlapped one another whilst, in the SCI cats, the second peak was shifted to the right to larger fiber areas and their equivalent fiber diameters (Figure 3B,C). The distribution, being bimodal, we replotted the data as cumulative histograms in Figure 4 where statistical analysis was performed with the Kolmogorov–Smirnov test [33].

The bimodal distributions for the nerves from control and experimental cats are evident with a visible break at ~40 µm^2^ in the cutaneous SUR nerve and ~50 µm^2^ in the SOL and MG muscle nerves (Figure 4). There was no change in the fiber areas or their equivalent fiber diameters in the SUR myelinated nerve fibers from the control cats or the experimental cats, 8 months after SCI (*p* > 0.05). There was also no significant change in the fiber areas for the small, myelinated nerve fibers (<40 µm^2^) of any of the nerves but there was a significant rightward shift of the fiber areas of >40 µm^2^ (and their equivalent fiber diameters) to large areas (and diameters) (*p* < 0.01; Figure 4). Not shown in this figure are the findings that there was no significant difference between the mean areas of SOL myelinated nerve fibers 3 weeks after SCI as compared to the mean areas of SOL nerve fibers 8 months after SCI, 77.8 + 15.6 µm^2^ (*n* = 3) and 91 + 8.6 µm^2^ (*n* = 7), respectively.

**Figure 3 biomedicines-10-02622-f003:**
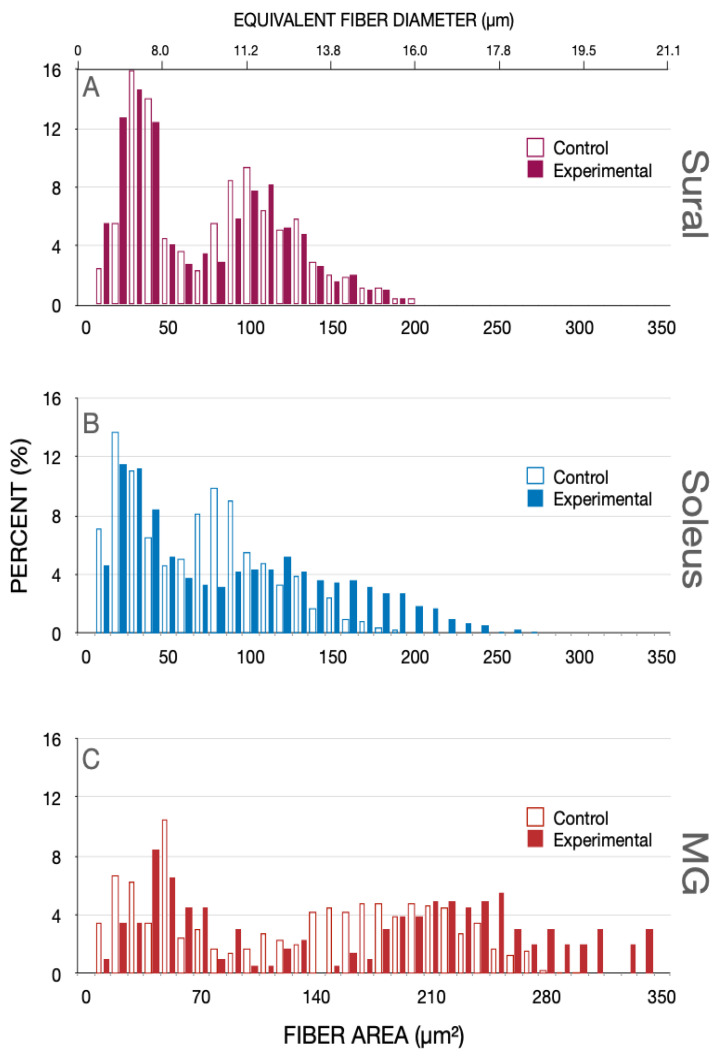
Percent histograms of the fiber areas measured in (**A**) sural (SUR), (**B**) soleus (SOL) and (**C**) medial gastrocnemius (MG) nerves that were removed from normal cats and cats in which spinal cord isolation was performed surgically, 8 months previously. The equivalent fiber diameter is shown on the upper *X*-axis of the plots in (**A**,**B**) Note the bimodal distributions of the nerve fiber areas and the equivalent fiber diameters of the sensory SUR nerve and the motor nerves of SOL and MG muscles. There was an obvious shift in the distributions of both SOL (**B**) and MG (**C**) nerves with the shift occurring only in the second larger population of the bimodal distributions. Because the distributions were bimodal, analysis of statistical significance of the shifts of the distributions to the right was performed on cumulative histograms shown in Figure 4.

**Figure 4 biomedicines-10-02622-f004:**
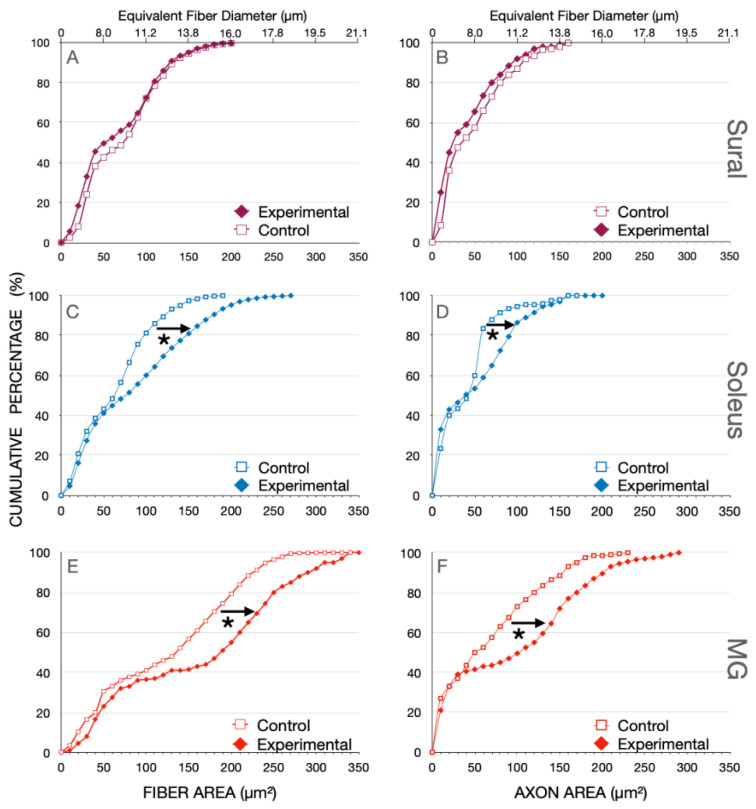
Cumulative percent histograms of the fiber and axon areas measured in (**A**,**B**) sural (SUR), (**C**,**D**) soleus (SOL), and (**E**,**F**) medial gastrocnemius (MG) nerves that were removed from normal cats and cats in which spinal cord isolation was performed surgically 8 months previously. Note that the cumulative distributions echo the bimodal distributions in both the pure sensory SUR nerve as well as in the SOL and MG motor nerves. The distributions of the SUR nerves were not different after SCI but those of the SOL and MG nerves were. Differences between the distributions were analyzed with the Kolmogorov–Smirnov test of significance (See methods). Statistically significant shifts of the distributions are denoted by * (*p* < 0.05).

### 3.2. Axon Size and Myelin Thickness

The significant shift to larger fiber and axon areas in the MG and SOL nerves after SCI (Figure 4) did not result in a change in the relationships between axon and fiber circumferences and between myelin thickness and the fiber circumference (Figure 5). Despite the myelinated nerve fibers of MG and SOL nerves appearing more circular after SCI than in the normal cats (Figure 2), the relationships of the axon and fiber circumferences (perimeters) were the same, with the slope of the two plots not being significantly different (cf the slopes ± SE being 0.78 ± 0.04 and 0.75 ± 0.05 with correlations coefficients of 0.94 in both cases; Figure 5A,B). In addition, the relationship between myelin thickness and the fiber circumferences was not altered in the silenced MG nerves after SCI (Figure 5C,D).

## 4. Discussion

This study of the effect of eliminating all neuromuscular activity in the hindlimb of cats demonstrates that the silenced myelinated motor nerve fibers significantly enlarge 8 months after spinal cord isolation (SCI), the transection of the spinal cord above and below the lumbosacral segments and cutting all the dorsal root nerve fibers that enter the isolated cord. The sensory fibers remained active because they were left in contact with their sense organs, including all the sural (SUR) myelinated afferent fibers conveying sensation from the skin and joints, and the ~40% of the medial gastrocnemius (MG) and soleus (SOL) myelinated afferent fibers with fiber areas of <50 µm^2^, emanating from the muscle spindles and joints. The size of these myelinated sensory fibers was unaffected by SCI, in stark contrast with the enlargement of the myelinated motor fibers innervating the MG and SOL muscles of the triceps surae group.

Because the normal relationship between the nerve fiber circumference and the myelin thickness of the fibers remained after SCI (Figure 5), the increase in size of the silenced myelinated motor fibers can be interpreted as a corresponding increase in their nerve conduction velocity, assuming the ratio of 6 m/sec and nerve fiber diameter in microns [36]. The diameters of the myelinated motor nerve fibers in cat hindlimbs correlate directly with their conduction velocities (correlation coefficient (r = 0–88, *p* < 0.001) [37]. In the experimental MG nerves after SCI surgery, the larger modal value was increased from 170 µm^2^ to 250 µm^2^, an increase of ~1.5× that represents the increase in size of the silenced MG motor nerves. Again, based on the assumption of the ratio of 6.0 m/sec and microns of fiber diameter, the increase in the size of the inactive nerve fibers represents an increase in nerve conduction velocity from 90 m/sec to 108 m/sec, a significant 20% increase in the nerve fiber conduction velocity. We make this assumption notwithstanding the considerable plasticity of the axon ion channel machinery including the voltage-gated sodium channel, NaV1.8 [38,39].

The mean (+ standard error) values of calculated conduction velocities of MG nerve fibers from the SCI cats in this study are compared in Figure 6 with those of the conduction velocities recorded in normally active cats and in cats in which the myelinated nerve fibers were electrically stimulated at 20 Hz daily in a 50% duty cycle (2.5 sec trains, repeated every 5 sec [19]). These comparisons highlight the differences between the size of myelinated motor nerve fibers that transmit action potentials during normal and high levels of activity, namely those that were electrically stimulated in the duty cycle of 50% per day, with a striking reduction in the size of the very active myelinated nerve fibers (Figure 6). In addition, the comparisons demonstrate the striking increase in size of the inactive nerve fibers in the SCI cats. It is important to note that the three diameters of the motor nerve fibers in this figure represent those of the motoneurons that are normally recruited into activity from small to large according to Henneman’s size principle [2,3,4]. The small nerves that innervate the non-fatiguing, slow contracting, low force generating motor units (MUs) are the most active, with the larger motoneurons innervating progressively more fatigable, fast contracting, and forceful muscle fibers, progressively recruited into activity as required for movement [4,6,40].

The nerve fiber diameters were measured after 8 months of SCI in this study. The question arises as to when this high asymptotic level was reached. We sampled only the SOL nerve at 3 weeks after SCI for measurement of the areas of the myelinated fibers. The finding that there was no significant change in the mean areas of SOL myelinated nerve fibers 3 weeks after SCI, in contrast to significant increase in the areas of the fibers at 8 months after SCI, indicate that the increase in the size of the silenced motor nerve fibers after SCI is a slow process. This is consistent with the findings of no change in size of the SOL and tibialis anterior motoneurons one month after SCI in cats [17] or after three months loading of cat plantaris muscle by removing triceps surae muscles [21]. The time scale in days in Figure 6 shows that the reduction in the conduction velocities of the 50% daily stimulated MG motor nerve fibers plateaued 50 days after initiation of the stimulation [19,41].

Our findings provide important evidence for a dynamic influence of neural activity and/or the size of the target muscle on nerve fiber caliber. The possibility that the size of the target muscle size has retrograde control of the size of motor nerve fibers is unlikely because SOL and MG motor nerves increased in size in our study of SCI (Figure 4C–F) despite atrophy of their corresponding muscles [22]. In addition, partial denervation that increased the numbers of muscle fibers innervated by each motoneuron, did not change the physiological parameters of nerve fiber size [9]. That neural activity changes the caliber of myelinated nerve fibers is supported by the evidence of the significantly slowing of MG nerve conduction velocities with chronic electrical stimulation at low frequencies for 50% of each day [19,41], together with conversion of fast-twitch muscles, their motor units, their motoneurons, and their muscle fibers to the slow type [19,40,41,42,43,44,45,46].

Our research supports that significant disruption of spinal cord input into the motoneurons increases the caliber of the motor nerve fibers and their axons. However, as we have not completed physiological measurements, we can only assume that this change supports an increase in conduction speed of these motor fibers to the levels of fast, fatigable motor units. Because our cats were all mature adults and not aged as they were in earlier studies [39,47], we discount the possibility that aging accounts for the changes in nerve size and conduction velocity that we document here. Considering research that has revealed considerable plasticity of the voltage-gated sodium channel isoforms in motor and sensory nerve membranes both with normal aging and during experimental trauma [38,39,47], we accept that our histological results need to be validated with investigations of membrane properties of the silenced motor nerves after SCI.

## 5. Conclusions

Our findings contribute to the data on plasticity of the neuromuscular system in its response to daily activity. This plasticity has been exploited in the training of athletes and has significance for general health and medical care. It is especially relevant for the growing elderly population and in the consideration of the care for patients recovering from surgeries that include repair of injured bone and/or peripheral nerves, long-stay hospitalized patients, the surgical management of cancer patients, and for spinal cord injured individuals.

## Figures and Tables

**Figure 2 biomedicines-10-02622-f002:**
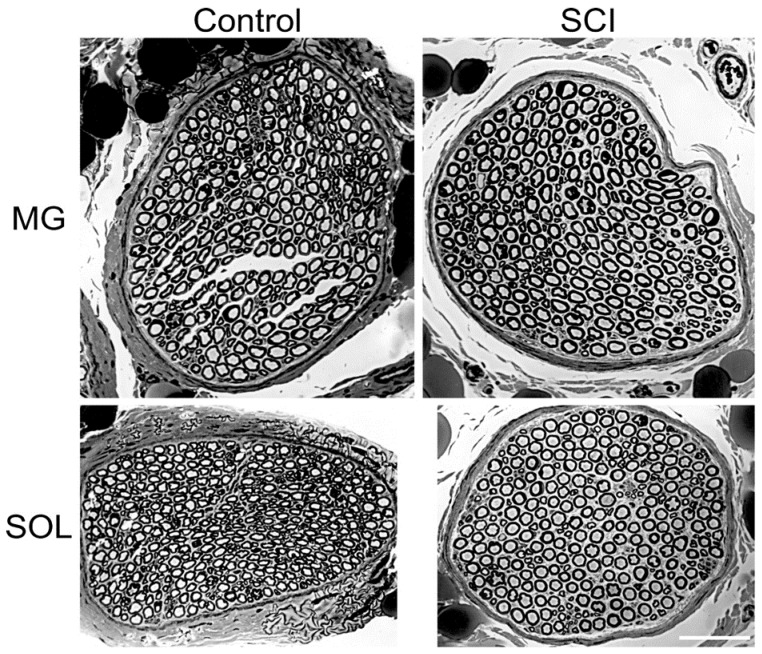
Photomicrographs of the cross-sections of the soleus (SOL) and medial gastrocnemius (MG) nerves that were dissected from normal cats and from cats in which the lumbosacral spinal cord was isolated [spinal cord isolation (SCI)] 8 months previously, as described in the Materials and Methods section. The scale bar is 50µm.

**Figure 5 biomedicines-10-02622-f005:**
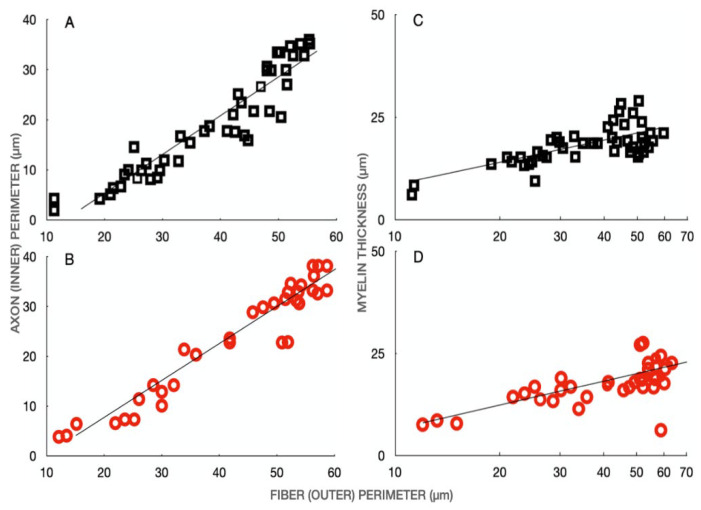
Regression analysis of the (**A**,**B**) calculated axon perimeters and (**C**,**D**) myelin thickness of the nerve fibers of medial gastrocnemius (MG) nerve as a function of the outer fiber perimeters from (**A**,**C**) normal cats (⬜), and (**B**,**D**) cats after spinal cord isolation (SCI) (⭕) 8 months previously. The slopes of the lines for the nerves from normal and SCI cats were not different, the slopes being (**A**) 0.78 +/− 0.04 (48) RO = 0.94 for the control MG nerve, (**B**) 0.75 +/− 0.05 (41) RO = 0.94 for the MG nerve after SCI, (**C**) 0.22 +/− 0.04 (48) RO = 0.63 for the control MG nerve and (**D**) 0.25 +/− 0.05 (41) RO = 0.66 for the MG nerve after SCI. Therefore, the g-ratios of inner and outer perimeters and hence, equal to the slopes of the regression lines, did not change with SCI.

**Figure 6 biomedicines-10-02622-f006:**
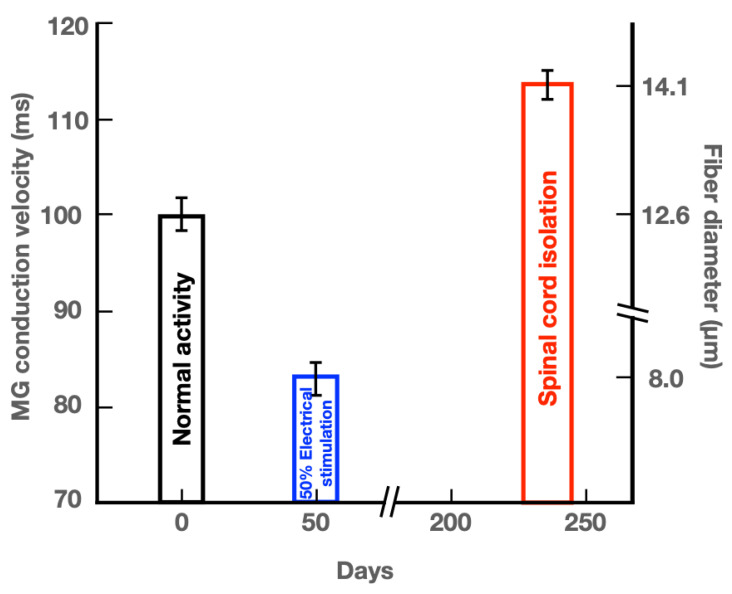
Histograms comparing the mean [± standard error (SE)] values of medial gastrocnemius (MG) nerve fiber diameters and conduction velocities in cats experiencing (1) normal neuromuscular activity, (2) 50% daily electrical stimulation (ES) in a 50% duty cycle, namely 2.5 sec on and 2.5 sec off for 5 days/week [19,40,41], and (3) little or no neuromuscular activity after spinal cord isolation (SCI) surgery. The size of motor nerve fibers (diameter) was calculated from recordings of single MG nerve fiber conduction velocities in hindlimbs of cats (1) experiencing normal daily levels of activity and (2) daily 50% duty cycles of ES. Fiber diameters are plotted on the right *Y* axis. The conduction velocities of inactive myelinated motor nerve fibers after SCI surgery were calculated from their measured diameters and the calculated mean (±SE) value of the calculated conduction velocity is plotted on the left *Y*-axis.

## Data Availability

All data are presented in the paper.

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
