# Peer review of "Enlargement of the Nerve Fibers of Silenced Lumbosacral Motoneurons in Cats"

_biomedicines, 2022, doi:10.3390/biomedicines10102622_

Round 1

Reviewer 1 Report

The manuscript comes from very experienced authors in the field of peripheral nervous system research. It brings supporting morphological evidence on the activity-dependent plasticity of the motor axons. It provides experimental data that 8-month following spinal cord isolation in cats, the myelinated motor nerve fiber size enlarge.

1)Only classical morphological data was extracted from 14 cats which is a very valuable material. Has other data been extracted form these experiments that has not been included here? Is this a re-analysis of a previously published material? This has to be very carefully detailed.

2)From the title and throughout the article it is specified “their nerve fibers enlarge”. Analysis is only performed on myelinated fibers (light microscopy). This should be emphasized throughout.

3)As no physiology was performed, the discussion on the “silenced motor nerve fibers” should be toned down. First of all, there are no field recordings to indicate the lack of spontaneous activity from the isolated spinal cord and no EMG recordings. Would the preserved sensory input not continue to drive the motoneurons in the segment? Furthermore, do the authors imply that soleus and medial gastrocnemius are purely motor? This has to be clarified.

4)The authors added a Fig.6 to discussion a physiology figure compiled from previous data and estimations. This should be made clearer in the legend. Changes in conduction velocity may not occur only due to changes in passible cable properties of the fibers. In fact, a lot of recent evidence suggests that motor axons are capable of a considerable plasticity of their ion channel machinery, i.e.  the voltage-gated Na+ channels isoforms like the NaV1.8. This should be at least discussed.

Author Response

Reviewer 1 Comments and Suggestions for Authors

The manuscript comes from very experienced authors in the field of peripheral nervous system research. It brings supporting morphological evidence on the activity-dependent plasticity of the motor axons. It provides experimental data that 8-month following spinal cord isolation in cats, the myelinated motor nerve fiber size enlarge.

  • Only classical morphological data was extracted from 14 cats which is a very valuable material. Has other data been extracted form these experiments that has not been included here? Is this a re-analysis of a previously published material? This has to be very carefully detailed.

Our response: Thank you for your kind acknowledgement of our experience. Regarding the question of whether other data has been extracted from the experiments, we apologize for this omission. When we wrote the manuscript, we could not locate Dr. Lynn Eldridge who had pioneered the technique of spinal cord isolation (SCI) at UCLA in California and who had performed the SCI surgeries, extracted, and fixed the nerves in 3% glutaraldehyde and sent them to my laboratory in Edmonton by rapid courier transport. A serendipitous encounter of author Dr. Aurora with a former neighbor of Dr. Eldridge in Canmore, Alberta in late August of this year allowed us to locate her in Salt Spring Island, British Columbia.  

Having located and communicated with Dr. Eldridge, we now include her in the authorship instead of the acknowledgements in Page 1 and we provide the information on methodology that was absent from the submitted manuscript. This includes a) reference in the INTRODUCTION, on page 3, paragraph 2, to an early paper that provided evidence of almost no detectable electromyographic activity in the atrophic hindlimb muscles but for the firing of a single moto unit detected in a few cat hindlimbs [26], b) evidence in the Materials and methods section on pages 5, paragraph 3, that all the lumbosacral motoneurons survived and retained their normal appearance after SCI surgery [27], c) details of the sterile surgeries to isolate the lumbosacral spinal cord (SCI) on pages 5 and 6 with the explanation of there being no official approval of the surgeries at UCLA in the absence of an Animal Care committee, d) information on page 7, paragraph 2, of recordings of muscle contractile properties made at 3 weeks and 8 months after the SCI surgeries and subsequent removal of muscle, spinal cord and peripheral nerves for investigation, e) description of the fixation of the nerves and their courtier transport to Edmonton for post-fixing in osmium tetroxide for cross-sectioning the nerves, and f) analysis and data interpretation, and writing of the manuscript by Drs Saljae and Gordon.   

.

  • From the title and throughout the article it is specified “their nerve fibers enlarge”. Analysis is only performed on myelinated fibers (light microscopy). This should be emphasized throughout.

Our response: Again, we thank this reviewer for pointing out this important point. We have emphasized that we are reporting on myelinated nerve fibers only. We refer to myelinated nerve fibers throughout the revised manuscript.

  • As no physiology was performed, the discussion on the “silenced motor nerve fibers” should be toned down. First of all, there are no field recordings to indicate the lack of spontaneous activity from the isolated spinal cord and no EMG recordings. Would the preserved sensory input not continue to drive the motoneurons in the segment? Furthermore, do the authors imply that soleus and medial gastrocnemius are purely motor? This has to be clarified.

Our response: We have addressed this concern as point a) of our response to Question 1.

  • The authors added a Fig.6 to discussion a physiology figure compiled from previous data and estimations. This should be made clearer in the legend. Changes in conduction velocity may not occur only due to changes in passible cable properties of the fibers. In fact, a lot of recent evidence suggests that motor axons are capable of a considerable plasticity of their ion channel machinery, i.e.  the voltage-gated Na+ channels isoforms like the NaV1.8. This should be at least discussed.

Our response: We have made an extensive revision of the legend to Figure 6 for clarification together with revising the related text in the last paragraph of page 15 and its continuation on page 16.

            We again thank the reviewer for pointing out the ion channel plasticity that we had failed to consider. We have included a sentence that refers to the NaV 1.8 channel in the second paragraph of page 15. We then argue in the paragraph which begins on page 17 and continues on page 18, that it is unlikely that aging accounts for our finding of increased size of silenced myelinated motor nerve fibers. Nonetheless, we acknowledge that ‘our histological results need to be validated with investigations of membrane properties of the silenced motor nerve after SCI.’    

Reviewer 2 Report

The manuscript by Gordon and Aurora focuses on the increase in size of nerve fibers following spinal cord isolation in cat hindlimbs below the L5-S3 dorsal root. The authors then measured the medial gastrocnemius (MG), soleus (SOL) and sural (SUR) nerves were removed for size measurements, eight months after SCI surgery and from age-matched control cats. Nerve fiber number, the linear relationship between axon size and myelin thickness, and the bimodal distributions of nerve fiber area and diameter were assessed after SCI. The SUR nerves were unchanged; whereas the MG and SOL nerves were increased in significantly in size. The authors conclude the SCI results in motor neuron size increases, as a dynamic response to neuromuscular activity and can adapt to injury.

The manuscript has an interesting approach to testing an unexplored area of motor neuron biology. However, there are many key experimental and technical details that are lacking. There are also several areas of clarification and data analysis that are needed to fully assess the findings showing increased nerve fiber sizes in the SCI stimulated cohorts. I would caution the authors to make sure to place their findings in proper context and make sure appropriate experimental details are provided.

Major Comments:

1. This sentence seems like it belongs in the “Discussion” and not the abstract “We discount that muscle atrophy had a retrograde influence on motoneuronal synthesis of neurofilaments because previous evidence demonstrated that high levels of neuromuscular activity induced by electrical stimulation reduced nerve conduction velocities and muscle fiber areas”. Please rephrase or remove from the abstract.

2. More details are needed on the animal experimental background in the Methods. Please state the age of the cats used, breed if known, and any relevant experimental details relevant to procuring the animals.

3. Weights and ages of the experimental cat cohorts are needed for reference.

4. Figure 2 is referenced in the results prior to Figure 1. Please reference Figure 1 prior to Figure 2.

5. Figure 3, please include either SEM or Std. Dev. for the percent calculations. Some statistics should be run if you are making the claim for larger fiber sizes.

6. Figure 4. Please state the statistical test used in the figure legend.

7. More details are needed on the SCI (not just the reference, 23). Please state how long exactly after SCI the cats were allowed to recover prior to euthanization? There is some detail in the results and Figure 6, but this need to be in the Methods as well.

8. Can the authors clarify if the electrically-stimulated cohorts increased muscle fiber sizes in the SCI cohorts as well?

Author Response

The manuscript focuses on the increase in size of nerve fibers following spinal cord isolation in cat hindlimbs below the L5-S3 dorsal root. The authors then measured the medial gastrocnemius (MG), soleus (SOL) and sural (SUR) nerves were removed for size measurements, eight months after SCI surgery and from age-matched control cats. Nerve fiber number, the linear relationship between axon size and myelin thickness, and the bimodal distributions of nerve fiber area and diameter were assessed after SCI. The SUR nerves were unchanged; whereas the MG and SOL nerves were increased in significantly in size. The authors conclude the SCI results in motor neuron size increases, as a dynamic response to neuromuscular activity and can adapt to injury.

The manuscript has an interesting approach to testing an unexplored area of motor neuron biology. However, there are many key experimental and technical details that are lacking. There are also several areas of clarification and data analysis that are needed to fully assess the findings showing increased nerve fiber sizes in the SCI stimulated cohorts. I would caution the authors to make sure to place their findings in proper context and make sure appropriate experimental details are provided

Our response: We thank this reviewer for these comments which we have considered carefully and made the requested revisions.

Major Comments:

  1. This sentence seems like it belongs in the “Discussion” and not the abstract “We discount that muscle atrophy had a retrograde influence on motoneuronal synthesis of neurofilaments because previous evidence demonstrated that high levels of neuromuscular activity induced by electrical stimulation reduced nerve conduction velocities and muscle fiber areas”. Please rephrase or remove from the abstract.

  Our response: We agree with this reviewer that this sentence does not belong in the abstract. We have deleted it from the abstract as requested.

  1. More details are needed on the animal experimental background in the Methods. Please state the age of the cats used, breed if known, and any relevant experimental details relevant to procuring the animals.

Our response: We have addressed this issue, providing more details of the animal experimental background, age, breed, and sex. In our response to the first reviewer, we detailed the many changes that we made in the MATERIALS AND METHODS section of the revised manuscript.

  1. Weights and ages of the experimental cat cohorts are needed for reference.

Our response: These are included under the heading of Animals on page 5 under the main heading of MATERIALS AND METHODS. We have now also included their description as long tailed cats precured from the Los Angeles and Ventura County Animal Shelters.

  1. Figure 2 is referenced in the results prior to Figure 1. Please reference Figure 1 prior to Figure 2.

 Our response: Thank you for pointing this out. We had numbered Figures 1 and 2 correctly in the results section but had not ordered the figure numbers correctly in the RESULTS section. We have made the appropriate correction to the first paragraph of the RESULTS section in page 10.

  1. Figure 3, please include either SEM or Std. Dev. for the percent calculations. Some statistics should be run if you are making the claim for larger fiber sizes.

 Our response: In answer to your recommendations we have added the sentence “The distribution, being bimodal, we replotted the data as cumulative histograms in Figure 4 where statistical analysis was performed with the Kolmogorov-Smirnov test [33].” to the last paragraph on page 10. We have also added the sentence ”Because the distributions were bimodal, analysis of statistical significance of the shifts of the distributions to the right was performed on cumulative histograms shown in Figure 4.” to the legend of Figure 3 on page 11.

  1. Figure 4. Please state the statistical test used in the figure legend.

Our response: I believe that we had stated the statistical test was the Kolmogorov-Smirnov test of significance in the figure legend of Figure 4 on page 13.

  1. More details are needed on the SCI (not just the reference, 23). Please state how long exactly after SCI the cats were allowed to recover prior to euthanization? There is some detail in the results and Figure 6, but this need to be in the Methods as well.

 Our response: We have now addressed the details on the SCI on pages 5 and 6 and stated on page 7 that the final experiments were performed 3 weeks and 8 months after the SCI surgery. With regard to how long after SCI the cats wee allowed to recover prior to euthanization,

  1. Can the authors clarify if the electrically-stimulated cohorts increased muscle fiber sizes in the SCI cohorts as well?

 Our response: We did not electrically stimulate any of the nerves in the control cats or in the experimental cats in which spinal isolation surgeries were performed. Perhaps our former discussion and presentation of our published data on electrical stimulation of nerves to the medial gastrocnemius in intact cat in Figure 6 (now on page 16 was confusing. We have made extensive revisions to the discussion in the last paragraph on page 15 and on page 16 to clarify our arguments for adaptation of nerve size to experimental changes in neuromuscular activity,    

Round 2

Reviewer 2 Report

The manuscript has been significantly improved since the original submission. Additional methodology and description of experimental details has been provided which is essential for reproducibility of the study findings. Extensive editing of the discussion and text are also an overall improvement. I have no additional edits or concerns about the manuscript.